# Pathophysiology and Clinical Meaning of Ventilation-Perfusion Mismatch in the Acute Respiratory Distress Syndrome

**DOI:** 10.3390/biology12010067

**Published:** 2022-12-30

**Authors:** Douglas Slobod, Anna Damia, Marco Leali, Elena Spinelli, Tommaso Mauri

**Affiliations:** 1Department of Anesthesia, Critical Care and Emergency, Fondazione Istituto di Ricovero e Cura a Carattere Scientifico Ca’ Granda, Ospedale Maggiore Policlinico, 20122 Milan, Italy; 2Department of Critical Care Medicine, McGill University, Montreal, QC H3A 3R1, Canada; 3Department of Pathophysiology and Transplantation, University of Milan, 20122 Milan, Italy

**Keywords:** electrical impedance tomography, acute respiratory distress syndrome, ventilation-induced lung injury, ventilation, perfusion

## Abstract

**Simple Summary:**

The balanced matching of air and blood flow in the lungs is crucial for maintaining oxygen levels and clearing carbon dioxide. In patients with severe respiratory failure due to acute respiratory distress syndrome, this matching is interrupted, precipitating life-threatening consequences. It has become increasingly evident that disturbances in the matching of air and blood flow may also contribute to the worsening of lung injury. In this review, we discuss classic and novel methods for measuring the adequacy of air and blood flow equilibration as well as clinical and experimental data that demonstrate a mechanistic link between mismatch and lung injury.

**Abstract:**

Acute respiratory distress syndrome (ARDS) remains an important clinical challenge with a mortality rate of 35–45%. It is being increasingly demonstrated that the improvement of outcomes requires a tailored, individualized approach to therapy, guided by a detailed understanding of each patient’s pathophysiology. In patients with ARDS, disturbances in the physiological matching of alveolar ventilation (V) and pulmonary perfusion (Q) (*V*/*Q* mismatch) are a hallmark derangement. The perfusion of collapsed or consolidated lung units gives rise to intrapulmonary shunting and arterial hypoxemia, whereas the ventilation of non-perfused lung zones increases physiological dead-space, which potentially necessitates increased ventilation to avoid hypercapnia. Beyond its impact on gas exchange, *V*/*Q* mismatch is a predictor of adverse outcomes in patients with ARDS; more recently, its role in ventilation-induced lung injury and worsening lung edema has been described. Innovations in bedside imaging technologies such as electrical impedance tomography readily allow clinicians to determine the regional distributions of V and Q, as well as the adequacy of their matching, providing new insights into the phenotyping, prognostication, and clinical management of patients with ARDS. The purpose of this review is to discuss the pathophysiology, identification, consequences, and treatment of *V*/*Q* mismatch in the setting of ARDS, employing experimental data from clinical and preclinical studies as support.

Acute respiratory distress syndrome (ARDS) is a heterogenous condition that is characterized by the development of inflammatory pulmonary edema and life-threatening hypoxemia, and it accounts for nearly 25% of patients who require mechanical ventilation [1]. The current Berlin definition of ARDS is based on the presence of bilateral pulmonary infiltrates upon thoracic imaging and a Pa_O2_/Fi_O2_ ≤ 300 mmHg during ventilation with a positive end-expiratory pressure (PEEP) of ≥5 cmH_2_O occurring within 7 days of exposure to a known clinical insult [2]. Congestive heart failure should not be the sole etiology. Aside from the implementation of “lung protective” ventilation strategies that lower tidal volume and lung inflation pressures [3], few interventions have been demonstrated to improve patient outcome in this condition, which has a mortality of 35–45% [1]. Recently, the notion that the way ARDS patients are ventilated can cause additional inflammation and damage to the injured lung, in a process referred to as ventilator-induced lung injury [4], has led to substantial research exploring the mechanisms determining and means to assess whether ventilation is optimal for a given patient.

## 1. Genesis of Ventilation-Perfusion Mismatch in the Acute Respiratory Distress Syndrome

Several pathophysiologic derangements give rise to gas exchange abnormalities in patients with ARDS.

### 1.1. Non-Ventilated Perfused Units (Shunt)

Arterial hypoxemia due to an intra-pulmonary shunt is a hallmark clinical problem. In 1979, Dantzker et al. used multiple inert gas elimination to study 16 patients with severe ARDS and demonstrated that nearly 50% of the cardiac output was distributed to pure shunt or very low *V*/*Q* lung zones, thus entirely explaining the observed hypoxemia [5]. The genesis of these non-ventilated lung zones is multifactorial. Non-aerated lung tissue may arise due to alveolar instability and collapse. Decreased surfactant production and abnormal pressure due to the superimposed weight of the edematous lung [6] contribute to alveolar instability and atelectasis, which have been associated with more severe lung injury and mortality [7]. These derangements are due, in part, to the inflammatory milieu caused by the primary etiologic factor, but mechanical ventilation alone is capable of decreasing surfactant levels in broncho-alveolar lavage fluid obtained from patients even in the absence of lung injury [8]. Such unstable lung units may be “re-opened” (recruited) with increasing airway pressure [9]. Lung units that remain non-aerated at higher levels of airway pressure have traditionally been considered consolidated as they occur due to inflammatory alveolar flooding with protein-rich fluid [10].

On the other hand, ventilation with a high inspired oxygen fraction can worsen shunt in ARDS patients by washing out alveolar nitrogen and promoting atelectasis. This occurs because nitrogen does not cross the alveolar–epithelial barrier and thus acts to maintain end-expiratory lung volume [11,12]. Regardless of the mechanism, the perfusion of poorly ventilated or non-ventilated lung zones is the main cause of arterial hypoxemia [5], and the degree of which is associated with the current definition of ARDS severity and clinical outcomes [1].

The impairment of vasoactive compensatory mechanisms also contributes to the extent of shunt. In normally functioning lungs, decreases in alveolar and/or mixed venous oxygen content trigger local vasoconstriction (hypoxic pulmonary vasoconstriction or HPV), which redirects blood flow towards ventilated lung zones, thereby decreasing functional shunting. Substantial experimental data have demonstrated that HPV is impaired in the setting of lung injury due to factors including endotoxemia [13] and the action of inflammatory mediators such as thromboxane and TNF-alpha [14,15].

Increased blood flow through intrapulmonary arterio-venous anastomoses contributes to shunt physiology. Such shunting is associated with a higher cardiac index and increased hospital mortality and is unrelated to PEEP level [16]. Intra-cardiac shunting through a patent foramen ovale is another cause of shunts in patients with ARDS, occurring in up to 20% of patients. Intra-cardiac shunting is associated with lower Pa_O2_/Fi_O2_, higher pulmonary artery pressure, and increased mortality [17]. It is also important to recall that decreased mixed venous oxygen content, due to reductions in cardiac output, can worsen hypoxemia when a significant concomitant intrapulmonary shunt is present.

### 1.2. Ventilated Non-Perfused Units (Dead-Space)

The other hallmark *V*/*Q* mismatch derangement in patients with ARDS is an increased fraction of physiological dead-space ventilation [18,19]. Patients with more severe lung injury demonstrate higher dead-space fractions [20]. Contributory causes include inflammatory intravascular thrombosis [21], with capillary microthrombi being more common in earlier ARDS stages and larger thrombi occurring throughout the disease course [22]. When present, these lesions are associated with the hypoperfusion of small pulmonary arteries [22]. The ventilation of non- or hypo-perfused lung regions results in impaired CO_2_ elimination and requires larger minute ventilation to compensate for hypercapnia. Hyperventilation can lead to the regional overdistension of non-dependent lung units. The compression of alveolar capillaries decreases regional perfusion, thereby generating high *V*/*Q* areas that directly contribute to increasing the global dead-space fraction and may also redistribute blood flow to non-aerated lung zones, which increases shunting (Figure 1).

Increased dead-space and markers of ventilatory inefficiency such as the ventilatory ratio (see Section 3.2) have also been associated with increased mortality in patients with ARDS [23,24].

In patients with ARDS due to COVID-19, specific pathophysiologic derangements of *V*/*Q* matching have been described, including increased dead-space caused by hypercoagulability and diffuse pulmonary angiopathy with impaired perfusion to ventilated lung zones [25].

The heterogeneity of the above *V*/*Q* mismatch profiles implies that some lung units will be non-ventilated and non-perfused. It is worth noting that although the matching of V and Q in these diseased lung units implies that functional shunt and dead space are not increased, these units are far from physiologic, and in fact, are associated with lung injury by various mechanisms, which are discussed below.

## 2. How to Assess *V*/*Q* Mismatch at the Bedside

### 2.1. Shunt at the Bedside

Few validated indices are available that can provide clinicians with a bedside estimate of a shunt fraction. As a result, shunt quantification is often neglected, despite the influence of shunt fractions on currently used dead-space indices [26]. The Berggren equation [27] is classically employed as follows:(1)$$\mathrm{Berggren}\;\mathrm{equation}:\;\frac{Q_{S}}{Q_{T}}=\frac{C_{cO_{2}}-C_{aO_{2}}}{C_{cO_{2}}-C_{vO_{2}}}$$
where $C_{cO_{2}}$, $C_{aO_{2}}$, and $C_{vO_{2}}$ are the capillary, arterial, and mixed venous oxygen content, respectively, and the resulting $Q_{S}/Q_{T}$ denotes venous admixture. This calculation corresponds to the shunt fraction in a classical Riley model [28]; however, it comprises both the flow from strictly shunted lung units and the “wasted flow” fraction from poorly ventilated (low $\dot{V}/Q$) units. $C_{cO_{2}}$ is usually estimated from the alveolar gas equation whereas $C_{aO_{2}}$ and $C_{vO_{2}}$ (or at least central venous content $C_{cvO_{2}}$) are measured.

As previously mentioned, the calculation of $Q_{S}/Q_{T}$ using the Berggren equation represents venous admixture, or the “physiological shunt”, which differs from the “true/anatomical shunt” in that it includes the contribution from low $\dot{V}/Q$ units. Berggren himself proposed that a 100% $F_{i}O_{2}$ trial could be used to eliminate $\dot{V}/Q$ inequalities after nitrogen washout and could thereby estimate the nature of true shunt [27]. It must be noted that the shunt fraction tends to increase with increasing $F_{i}O_{2}$, possibly due to oxygen-induced atelectasis and/or a redistribution of pulmonary blood flow [12], and this method has been criticized [29,30]. However, its usefulness in predicting anatomic shunts has recently been demonstrated using imaging studies [31]. Given that pulmonary artery catheterization is not frequently performed in ARDS patients, the applicability of the classic calculated venous admixture is limited. However, methods that do not require mixed venous sampling have been proposed [32].

As stated above, the definition of physiological shunt is overly simplistic, as it lumps the effects of a true shunt, *V*/*Q* inequalities, and oxygen diffusion impairment. The relationship between the fraction of inspired oxygen and arterial oxygen saturation may be more informative and can be adequately explained by more refined lumped-parameter models that separate a true shunt from *V*/*Q* mismatch and/or diffusion impairment [33,34,35]. Such models have been derived by combining arterial blood gas analyses with noninvasive monitoring and expired gas analysis during the sequential administration of varying fractions of inspired oxygen, which constitutes a procedure lasting 10 to 15 min that can be performed at the bedside. This system is known as the Automatic Lung Parameter Estimator (ALPE) [33] and provides a valuable tool with which to estimate shunts without the need for pulmonary artery catheterization.

Finally, echocardiography can be used at the bedside to diagnose intra-cardiac shunting through a patent foramen ovale (PFO), which is a potential contributor to hypoxemia in ARDS patients.

### 2.2. Dead-Space at the Bedside

Volumetric capnography provides a noninvasive method for estimating physiological dead-space. By assuming that the volume of CO_2_ exhaled per breath ($V_{ECO_{2}}$) equals the volume of CO_2_ cleared by the portion of ideally perfused and ventilated alveoli ($V_{ACO_{2}}$), and that tidal volume is the sum of air reaching these alveoli plus that wasted in the physiological dead-space, ($V_{T}=V_{T-ALV}+V_{D-PHYSIO}$), an equation can be derived to calculate the physiological dead-space fraction:(2)$$\mathrm{Physiological}\;\mathrm{dead}-\mathrm{space}\;\mathrm{fraction}:\;\frac{V_{D-PHYSIO}}{V_{T}}=\frac{P_{\bar{AI}CO_{2}}-P_{\bar{E}CO_{2}}}{P_{\bar{AI}CO_{2}}}$$

$V_{T}$ is the tidal volume, $V_{D-PHYSIO}$ is the physiological dead-space volume, and $P_{\bar{E}CO_{2}}$ and $P_{\bar{AI}CO_{2}}$ are the mean CO_2_ tensions in the whole exhaled breath and in the ideally ventilated and perfused alveoli (*V*/*Q* = 1), respectively [36]. The drawback of this approach is that $P_{\bar{AI}CO_{2}}$ cannot be measured. By equating it with the mean $P_{CO_{2}}$ of alveolar air, the Bohr equation [37] underestimates dead-space, whereas the modified Bohr–Enghoff equation [38] overestimates it by approximating $P_{\bar{AI}CO_{2}}$ to PaCO_2_, thus disregarding the effect of shunt on arterial CO_2_ [26,39]. However, these equations allow for a practical bedside estimation. Notably, these calculations assume a steady state of alveolar ventilation, perfusion, and CO_2_ production [36].

Other features of the volumetric capnogram [40], including the phase III slope (S_III_), provide an estimate of *V*/*Q* heterogeneity among alveolar units. This has been validated against the multiple inert gas elimination technique (MIGET) applied to pigs [41], while in humans it showed a significant, though weak, correlation with severity assessed by the lung injury score [42]. The slope estimation of capnograms of varying morphology allows for breath-by-breath monitoring but is not straightforward, which limits the accuracy of this technique [43].

Unfortunately, the use of volumetric capnography has been limited in intensive care units (ICUs) so far. To perform secondary analyses of ARDS trials, non-capnographic estimates of dead-space have been validated. These rely on empirical estimates of resting energy expenditure (REE) and VCO_2_ [44]. When compared to measured Bohr–Enghoff dead-space, the least biased among them might be a technique based on the Harris–Benedict equation [44,45], but agreement between the two remains quite poor [46].

A simple bedside Index reflecting *V*/*Q* inequalities is the ventilatory ratio (VR). It makes many of the same assumptions as the Bohr–Enghoff equation and is inversely correlated with the efficiency of ventilation; additionally, it does not require capnography or REE estimates but relies only on arterial blood sampling and predicted minute ventilation and PaCO_2_ in healthy subjects [47]. Its good correlation with dead-space ventilation has been confirmed both by computational and clinical studies [48,49]; however, regarding physiological dead-space, the relative contribution of a shunt is not accounted for.

The use of the ratio between end-tidal and arterial CO_2_ tension ($P_{ETCO_{2}}/P_{aCO_{2}}$) has been proposed by Gattinoni and colleagues as a measure of lung performance as a “gas exchanger” [50,51] and has been validated retrospectively [51,52] as a marker of ARDS severity. Since it only uses end-tidal capnography and arterial blood sampling, it is a useful tool for detecting gross *V*/*Q* inequalities. Yet, it suffers from the same limitations as previous indices, failing to clearly separate the relative contributions of dead-space and shunts [50,51,52].

### 2.3. Novel Techniques

Recent advancements may provide means for assessing *V*/*Q* mismatch at the bedside, with the potential for separating the relative contribution of a shunt and dead-space. Interest in thoracic electrical impedance tomography (EIT) is increasing due to its ability to noninvasively measure the regional distribution of ventilation in real time [53,54]. EIT functions by producing low-voltage, high-frequency alternating currents (e.g., 5 mA, 50 kHz) [53] sequentially across a set of electrodes, commonly via a 16-electrode thoracic belt. Typically, two electrodes provide the current, while the other non-stimulating electrodes record voltages. An inverse problem is then solved such that changes in background conductivity within the thorax due to tidal ventilation are reconstructed [55].

More recently, EIT monitoring has allowed for the determination of regional lung perfusion via the modeling of the first-pass kinetics of a central venous bolus of a 5% saline bolus under apneic conditions [56]. This involves separating the cardiac and lung compartments through which the saline bolus flows and then using the maximal slope method to estimate regional blood flow in the lung regions. Further detail can be found in the publications by Borges et al. [56] and Kircher et al. [57]. This method has been validated against computed tomography (CT) scans, single proton emission computerized tomography (SPECT), and positron emission tomography imaging [56,58]. By superimposing ventilation and perfusion maps, the number of unmatched units can be measured [59,60,61]. Although EIT only samples a portion of the thorax [62], it provides regional ventilation and perfusion data that are representative of the whole lung, separates between high and low *V*/*Q* units, and provides data that can be used to derive continuous *V*/*Q* distributions in a MIGET-like fashion [63].

Finally, model-based approaches are emerging. Automated data collection while varying FiO_2_ allows for an estimation of shunt and dead-space [33], while pulse oximetry coupled with molecular flow-sensing (MFS) of exhaled gases can be modelled to obtain accurate continuous *V*/*Q* distributions [64].

In the future, a wealth of information is expected to be available at the bedside of patients in the ICU. This will require adequate consideration of a given tool’s potential to precisely distinguish shunts from dead-space and interpret their clinical significance.

## 3. *V*/*Q* Mismatch as a Marker of Severity in ARDS Patients

### 3.1. Intrapulmonary Shunt

Although the shunt fraction contributes to the important clinical problem of arterial hypoxemia, its role per se as a potential marker of ARDS severity and adverse outcomes is less established. A modification of the Berggren equation has been related to the risk of developing ARDS in critically ill patients and of increased right ventricular workload [65]. Importantly, the shunt fraction influences dead-space indices [26] and new technologies might increase the feasibility of shunt estimation at the bedside.

### 3.2. Dead-Space

The Bohr–Enghoff calculation of physiological dead-space has arguably provided one of the strongest indicators of mortality in ARDS patients. The dead-space fraction as determined by volumetric capnography early in the course of ARDS predicts mortality independently of oxygenation, the Simplified Acute Physiology Score II (SAPS-II), and vasopressors [23,66,67]. In the study by Nuckton and colleagues, every 0.05 (i.e., 5%) increase in the dead-space fraction, which was measured a median of 11 h after ARDS diagnosis, was independently associated with a 45% increase in mortality odds [23]. In addition, serial measurements of physiological V_D_/V_T_ by volumetric capnography have been shown to identify patients at increased risk of death up to 6 days after meeting ARDS criteria [66,67]. The use of the threshold value of V_D_/V_T_ as a prognostic tool may differ among different ARDS etiologies, with aspiration and infectious pneumonia having significantly higher V_D_/V_T_ than non-pulmonary sepsis or trauma. Regardless of etiology, V_D_/V_T_ was significantly increased in non-survivors versus survivors [68]. Recently, the prognostic value of physiological dead-space has also been studied in critically ill COVID-19 ARDS patients, showing an interesting association with coagulation. Increased V_D_/V_T_ was associated with higher D-dimer levels and a lower likelihood of being discharged alive. A V_D_/V_T_ above 57% was used to help identify a high-risk subgroup of patients independent of the Pa_O2_/Fi_O2_ ratio [69].

Dead-space estimation based on resting energy expenditure has been proposed as a bedside index capable of predicting mortality based on secondary analyses of two prospective studies [70] but other analyses have failed to demonstrate improved outcome prediction when added to measures of oxygenation and respiratory mechanics [71,72]. Nevertheless, the estimated and measured dead-space fractions had a similar ability to predict the extent to which extracorporeal CO_2_ removal reduced driving pressure (DP) when ultra-protective ventilation was applied, but only the measured dead-space was associated with mortality [46]. Of note, compared to other empirical dead-space formulae, a direct estimate based on the least angle regression of physiological variables proposed by Beitler and colleagues (direct estimation) seems relatively less biased and may retain some potential for prognostication [44,72].

The ventilatory ratio is an index that relates a patient’s measured minute ventilation and PaCO_2_ to the predicted minute ventilation required to achieve a PaCO_2_ of 37.5 mmHg.
(3)$$\mathrm{Ventilatory}\;\mathrm{ratio}:\;\frac{\mathrm{Minute}\;\mathrm{ventilation}\;\left(\mathrm{mL}/\min\right)\times {\;\mathrm{Pa}}_{\mathrm{CO}2}\;\left(\mathrm{mmHg}\right)}{\mathrm{Predicted}\;\mathrm{body}\;\mathrm{weight}\;\left(\mathrm{kg}\right)\times 100\;\left(\mathrm{mL}/\mathrm{kg}/\min\right)\times 37.5\;\left(\mathrm{mmHg}\right)}$$

This is an easy-to-use bedside calculation that operates as an outcome predictor in a way that is strikingly similar to the measured physiological dead-space fraction. Higher values indicate greater ventilatory inefficiency and a higher dead-space fraction [47]. It has been shown to be capable of predicting mortality independently of oxygenation, shock status, vasopressor use, and Acute Physiology and Chronic Health Evaluation III (APACHE-III) scores [24,49,73], with odds ratios comparable to those of Nuckton and colleagues [24], thereby outperforming empirical estimates, except with respect to direct estimation [72]. The trajectory of ventilatory ratios during the early stages of ARDS has also been associated with survival and may prove valuable when tracking a patient’s clinical course [72,74].

Interestingly, in a recent secondary analysis of the large (*n* = 927) PRoVENT-COVID study cohort, none of the previously discussed formulae for dead-space estimation, nor the ventilatory ratio, were found to be independent predictors of mortality in an adjusted base risk model. Only direct estimation—when measured early at the start of ventilation, but not on the following day—retained additional prognostic value. The authors also argued that in patients with COVID-19, dead-space might be regarded more as a marker of ARDS severity rather than as an independent prognostication factor. Despite its apparent simplicity, the P_ET_CO_2_/PaCO_2_ ratio was the only index in this cohort to be independently associated with outcomes both at the start of ventilation and on day one [71] and has been recently validated as a predictor of mortality in a large retrospective cohort [52].

### 3.3. Assessment of V/Q Mismatch by EIT

The sum of the percentages of non-perfused ventilated and perfused non-ventilated lung units (unmatched units) obtained by superimposing EIT-derived ventilation and perfusion maps has recently been shown to be capable of independently predicting mortality in ARDS patients. A value of 27% unmatched units predicted mortality with a positive predictive value of 67% and a negative predictive value of 91%. The percentage of only perfused units (i.e., an estimate of a shunt) was significantly inversely correlated with the Pa_O2_/Fi_O2_ ratio and the dorsal fraction of ventilation [61]. These findings represent an additional means to assess ARDS severity at the bedside and lay the foundation for more advanced analyses. EIT imaging offers the potential to differentiate between dead-space and shunt fractions, potentially bridging the gap between bedside measures and respiratory pathophysiology.

## 4. Hypoxic Pulmonary Vasoconstriction and *V*/*Q* Mismatch as Mechanisms of VILI

Beyond causing impairments in gas exchange, *V*/*Q* mismatch and the physiologic responses to mismatch have been implicated in the development and progression of lung injury in patients with ARDS. A summary of the corresponding explanatory mechanisms is presented in Table 1.

### 4.1. VILI Related to Perfused Non-Ventilated Lung Units

The perfusion of low *V*/*Q* and shunted lung zones in healthy lungs is counterbalanced by hypoxic pulmonary vasoconstriction (HPV), a physiological response to alveolar hypoxia and/or to decreased mixed venous oxygen content that diverts pulmonary blood flow from poorly ventilated to more normally ventilated lung zones [90], thereby improving *V*/*Q* matching and gas exchange. HPV has been extensively investigated in experimental studies. In ARDS models induced by E. Coli endotoxin and oleic acid infusion, HPV appeared to be inhibited [75,76,91,92], contributing to increased shunt and worsening oxygenation. In other models, induced mainly by oleic acid infusion, HPV was preserved [77]. In ARDS patients, the effectiveness of HPV might vary depending on etiology, hemodynamic status, the administration of medications, and preexisting pulmonary conditions. However, both experimental [78] and clinical [79] observations of worsening acute pulmonary hypertension induced by hypoxemia and acidosis, and of the hypoxemic effects of intravenous vasodilators, suggest that HPV is preserved in most ARDS patients.

The observation that intrapulmonary shunting and physiological HPV redistribute pulmonary perfusion has shed light on the potentially deleterious effects of hypo-perfused lung zones in the genesis and/or progression of lung injury in ARDS. Studies of healthy lungs demonstrated that HPV optimizes gas exchange but may also limit the supply of oxygen and nutrients to shunted lung zones [80]. In the presence of atelectasis, the size of the aerated lung is decreased, thus increasing the risk of overdistension and barotrauma of the ventilated areas and, consequently, promoting inflammation [81]. SPECT analysis of pigs undergoing one-lung ventilation (OLV) for 90 min followed by two-lung ventilation (TLV) showed hyperinflation and hyperperfusion of the ventilated lung, which caused diffuse damage to the alveolar compartment [82]. However, most studies of OLV models in the literature were performed for only a few hours and focused on the development of lung injury after TLV was restored (ischemia-reperfusion injury). Recently, the impact of OLV with a higher and lower V_T_ for 24 h without restoration of TLV was studied in pigs by our group [83] (Figure 2).

OLV caused bilateral lung injury and, interestingly, lowering V_T_ prevented injury to the ventilated lung, but this protection was only partial in the non-ventilated lung. EIT analyses demonstrated that lung stress (classical VILI mechanism) was the main mechanism of injury in the ventilated lungs but collapse and hypoperfusion were implicated in the non-ventilated lungs. Notably, inflammation measured in the non-ventilated lung was dampened by reducing V_T_ to the contralateral ventilated lung, indicating the possible role of inflammation-based crosstalk between the lungs.

### 4.2. VILI Related to Ventilated Non-Perfused Lung Units

The other extreme of *V*/*Q* mismatch is represented by areas with high and infinite *V*/*Q* (dead-space), which have been studied in experimental settings mostly by intravascular occlusion or the surgical ligation of the pulmonary arteries. Alveolar hypocapnia in ventilated non-perfused alveoli seems to be responsible for local lung injury, which is, in part, mediated by alterations of the surfactant system [85] that lead to alveolar damage due to instability [86,87,88,89,93,94] and apoptosis [95]. Non-perfused ventilated areas can develop hemorrhagic infarction, as shown in a model of cardiopulmonary bypass and preserved ventilation [96]. Healthy swine lungs undergoing regional pulmonary vascular occlusion were studied to assess the local diversion of V_T_ from nonperfused to perfused areas via computed tomography. This diversion of ventilation appeared to be a compensatory mechanism that effectively limits *V*/*Q* mismatch [97] but also an indirect mechanism of lung injury due to overdistention and injury to perfused ventilated regions (Table 1 and Figure 2, panel C). As with HPV for perfusion, hypocapnic bronchoconstriction can divert ventilation, improve *V*/*Q* matching, and optimize gas exchange by redistributing V_T_ to better-perfused lung areas [98,99]; on the other hand, it may contribute to the development of injury in these areas due to concomitant hyperventilation and hyperperfusion [100]. In a model of pulmonary hypertension induced by E. Coli endotoxin in sheep, it was shown that there is no threshold for edema formation when capillary permeability was increased; any increase in pulmonary blood flow or pressure increased edema [101]. This finding supports the hypothesis that an increase in regional lung perfusion in conditions of local injury (for example, due to increased lung stress from hyperventilation) can result in further injury due to edema formation. Furthermore, mild bilateral lung injury that develops in dogs after unilateral pulmonary artery occlusion is characterized by endothelial abnormalities and perivascular edema [102]. Broccard et al. observed that the dependent distribution of VILI in supine large animals ventilated with high V_T_ might be explained by regional differences in blood flow and vascular pressure, suggesting that differences among ventilatory patterns may be due, at least partially, to differences in hemodynamics [103].

Since the 1960s, the role of alveolar hypocapnia in the development of lung injury in non-perfused ventilated lung units has been demonstrated by studying the impact of inhaled CO_2_ in unilateral pulmonary artery ligation (UPAL) models. Edmunds et al. first found a significant reduction in atelectasis in the ligated lung and a local increase in ventilation induced by inhaled CO_2_, which led them to hypothesize that there was a direct effect of CO_2_ or [H+] on bronchiolar alveolar cells and surfactant [104]. Kolobow et al. observed a reduction in hemorrhagic infarction and decreased alveolar and capillary injury in spontaneously breathing lambs during total cardiopulmonary bypass coupled with inhaled CO_2_. The effect of high PCO_2_ inhalation was also studied in preterm lambs; the result was an increase in lung gas volume and a reduction in histological damage and inflammation [105]. Recently, our group described a significant reduction in bilateral lung injury due to left-sided UPAL by administering 5% inhaled CO_2_ during controlled mechanical ventilation [84] (Figure 2). Our findings confirmed the protection of the ligated lung but also highlighted the protective role of inhaled CO_2_ in the non-ligated lung, which was less overdistended due to a more homogenous distribution of ventilation as assessed by EIT. Our group also investigated the question of whether the protective effects of inhaled CO_2_ in the setting of bilateral lung injury caused by UPAL were due to increased PaCO_2_ or to the local effects of CO_2_ inhalation. We demonstrated that inhaled CO_2_ allows for more effective bilateral lung protection compared to plasmatic hypercapnia obtained through other methods [106].

In summary, in the presence of an elevated dead-space fraction, ventilated non-perfused units can be damaged by the inhibition of surfactant production and function, the induction of apoptosis, local ischemia, and inflammatory crosstalk from the residual hyperventilated and hyperperfused lung.

## 5. Strategies for Decreasing *V*/*Q* Mismatch in ARDS Patients

Clinical strategies that can improve *V*/*Q* matching in ARDS patients include the administration of intravenous or inhaled vasoactive agents, the optimization of ventilation settings, and the implementation of prone positioning (Figure 3).

Interventions that redistribute perfusion away from non-ventilated lung zones towards well-ventilated zones reduce intrapulmonary shunting and improve oxygenation. Likewise, the restoration of perfusion to ventilated but non-perfused lung zones reduces alveolar dead-space, thereby increasing ventilatory efficiency. In addition to improved gas exchange, and, possibly more importantly, due to the prevention of the abovementioned mechanisms, reducing shunt and dead-space may lead to more effective lung-protective ventilation.

### 5.1. Vasoactive Drugs

Decades of clinical research have demonstrated that gas exchange can be improved by pharmacologically manipulating *V*/*Q* distribution. Intravenous almitrine, a selective pulmonary vasoconstrictor, enhances HPV by enhancing the redistribution of blood away from un-ventilated lung zones, whereas inhaled nitric oxide (iNO) is a selective pulmonary vasodilator that preferentially increases perfusion to well-ventilated lung zones.

In 1987, Reyes et al. used MIGET to study nine patients with ARDS before and during the administration of almitrine [107]. Almitrine reduced Qs/Qt from 29 ± 11 to 17 ± 11% and increased perfusion to normal *V*/*Q* lung zones (63 ± 9 to 73 ± 6%), resulting in an increase in Pa_O2_ from 78 ± 15 to 138 ± 52 mmHg. Given alone, almitrine increased pulmonary artery pressure and—if infusion was protracted—the development of non-ventilated, poorly perfused regions could worsen hypoxemia.

In 1998, Gallart et al. administered intravenous almitrine and iNO alone and together to 48 patients with ARDS [108]. When almitrine was given alone, they also reported a decrease in Qs/Qt compared to the baseline (38 ± 1 to 33 ± 1%) and found that the concomitant administration of iNO led to a further reduction in Qs/Qt (30 ±1%) and maintained pulmonary artery pressure at the baseline value.

Rossaint et al. administered iNO to 10 patients with severe ARDS and used MIGET to demonstrate a reduction in intrapulmonary shunting from 36 ± 5 to 31 ± 5% (*p* = 0.028) with an improvement in Pa_O2_/Fi_O2_ [109].

More recently, Wang and Zhong used EIT perfusion imaging in a severe ARDS patient with a ventral predominant distribution of ventilation to demonstrate that iNO led to a regional redistribution of perfusion to the ventral lung, decreasing the percentage of only-perfused lung units from 14% to 9% and dramatically increasing the Pa_O2_/Fi_O2_ [32].

In patients with severe ARDS due to COVID-19, Bagate et al. administered iNO, alone and in combination with intravenous almitrine to patients ventilated in the supine position and compared these interventions to prone positioning without vasoactive therapy [110]. Interestingly, the greatest improvement in Pa_O2_/Fi_O2_ was observed during combined therapy with iNO and almitrine, which exceeded the increase in oxygenation observed during prone positioning, suggesting that impairments in *V*/*Q* matching might be distinct in patients with COVID-19.

Despite the demonstrated improvements in gas exchange, neither drug has been associated with a mortality benefit in patients with ARDS [111], perhaps due to the further reduction in perfusion to non-ventilated ischemic regions.

### 5.2. Positive End-Expiratory Pressure and Recruitment

In mechanically ventilated patients with ARDS, the appropriate setting of positive end-expiratory pressure (PEEP) may stabilize the re-opening of perfused, atelectatic lung units and decrease shunting. Excessive PEEP is associated with an overdistension of lung units, which increases dead-space and redistributes perfusion to unventilated lung zones, thereby increasing shunting [112]. Although several large, international, randomized-controlled trials have failed to demonstrate a mortality benefit associated with a particular strategy for the setting of PEEP [113,114,115,116], many studies have illustrated the benefits of PEEP titration towards *V*/*Q* matching.

In 1985, Ralph et al. studied 16 ARDS patients with MIGET during a PEEP trial [117]. PEEP-induced increases in PaO_2_ of >10 mmHg compared to the baseline were associated with a reduction in shunt and perfusion to low *V*/*Q* lung zones. Occasionally, higher PEEP increased the ventilation of lung zones, with *V*/*Q* ratios > 10, which was suggestive of alveolar overdistension.

More recently, Karbing et al. performed a combined CT scan and *V*/*Q* analysis on 12 patients with ARDS, studied at a PEEP of 5 followed by 15 or 20 cmH_2_O [118]. *V*/*Q* mismatch was evaluated using the ALPE method, which relies on measurements of end-tidal and blood gas analyses. A higher PEEP resulted in a decrease in intrapulmonary shunting, which was correlated with an increase in normally aerated lung tissue visualized via CT scan. Interestingly, in a subset of four patients, both shunting and high *V*/*Q* increased at higher PEEP, suggesting a predominant alveolar overdistension effect.

Perier et al. used EIT perfusion monitoring to study nine patients with ARDS secondary to COVID-19 at a PEEP of 6, 12, and 18 cmH_2_O [119]. Higher PEEP was associated with increasing oxygenation but a decrease in lung and respiratory system compliance. Higher PEEP resulted in a decrease in the ventral distribution of ventilation, suggesting regional overdistension, and this was accompanied by a decrease in ventral perfusion. In the dependent lung, a higher PEEP was associated with a reduction in dorsal EIT-measured shunt.

We previously studied 10 patients with ARDS secondary to COVID-19 using EIT monitoring at a PEEP of 5 and 15 cmH_2_O. Higher PEEP increased the Pa_O2_/Fi_O2_ and the distribution of ventilation to the dependent lung, suggesting alveolar recruitment [59]. However, higher PEEP also resulted in a higher PaCO_2_ and ventilatory ratio, suggesting a concomitant increase in dead-space ventilation.

Later, we used EIT perfusion imaging to study regional *V*/*Q* matching in 15 patients with ARDS at 2 PEEP levels (5 and 15 cmH_2_O) [63] (Figure 3). This population was characterized by a greater potential for lung recruitment based on a range of EIT-based R/I ratios of 0.62–2.67. EIT demonstrated that higher PEEP was associated with a reduction in wasted ventilation (the ventilation of lung zones with a *V*/*Q* ratio > 1) in the non-dependent lung and a decrease in wasted perfusion (the perfusion of lung zones with a *V*/*Q* ratio < 1) in the dependent lung. Interestingly, and in a manner that potentially unifies previous findings, the improvement of *V*/*Q* mismatch was correlated with recruitability, further highlighting the rationale for the personalization of PEEP based on this characteristic. An advanced analysis of the EIT perfusion data using a “MIGET-like” approach demonstrated that higher PEEP shifted the average distributions of ventilation and perfusion (mean V and mean Q) towards a *V*/*Q* ratio of 1 in the non-dependent lung.

### 5.3. Prone Positioning

Prone positioning is a clinical intervention associated with improvements in *V*/*Q* matching and mortality [120]. In the Perier study, prone positioning caused a dependent redistribution of ventilation [119] whereas perfusion was predominantly dorsally distributed, and the distribution was unaffected by prone positioning. This resulted in a reduction in shunting and perfusion to low *V*/*Q* lung zones in the dependent lung and an improvement in Pa_O2_/Fi_O2_.

In 2022, we studied 21 patients with ARDS secondary to COVID-19 during ventilation in the supine and prone positions using CT scans and EIT perfusion monitoring [60] (Figure 3). Prone positioning resulted in dorsal lung recruitment measured via CT scan, increased the Pa_O2_/Fi_O2_ ratio, and decreased the measured venous admixture. EIT demonstrated a non-dependent decrease in only-ventilated lung units and a decrease in the dead-space to shunt ratio, suggesting an improvement in *V*/*Q* matching.

Prone positioning has also been demonstrated to increase the ratio of minute ventilation divided by PaCO_2_, suggesting decreased dead-space. In a prospective study of 225 patients with ARDS, an increase in this ratio after prone positioning was associated with a reduction in mortality [121]. Interestingly, improved oxygenation was not associated with any mortality benefit.

In intubated, spontaneously breathing patients with COVID-19, Pierrakos et al. demonstrated that prone positioning improved ventilatory homogeneity and increased dorsal respiratory system compliance, which were measured using EIT [122]. There was no change in Pa_O2_/Fi_O2_ and the observed benefits in mechanics disappeared soon after the patients’ resumption of a supine position.

Preliminary data have also demonstrated a physiologic benefit provided by prone positioning in the setting of unilateral pulmonary consolidations and severe hypoxemia [123]. In this setting, a more homogeneous distribution of ventilation in the healthy lung during prone positioning favored a redistribution of perfusion from the affected lung to the unaffected lung and decreased the percentage of non-ventilated perfused lung units in the consolidated lung, thereby decreasing the calculated venous admixture and improving gas exchange.

## 6. Conclusions

Ventilation–perfusion mismatch is generated by the very same mechanisms that determine the severity of ARDS (collapse, overdistension, vascular abnormalities, and altered regional mechanics). In addition, the presence of *V*/*Q* mismatch (both shunting and dead-space predominant phenotypes) can aggravate lung injury. For these reasons, *V*/*Q* mismatch may represent an accurate marker of ARDS outcome and a useful tool with which to guide personalized interventions, such as setting higher PEEP in more recruitable patients.

## Figures and Tables

**Figure 1 biology-12-00067-f001:**
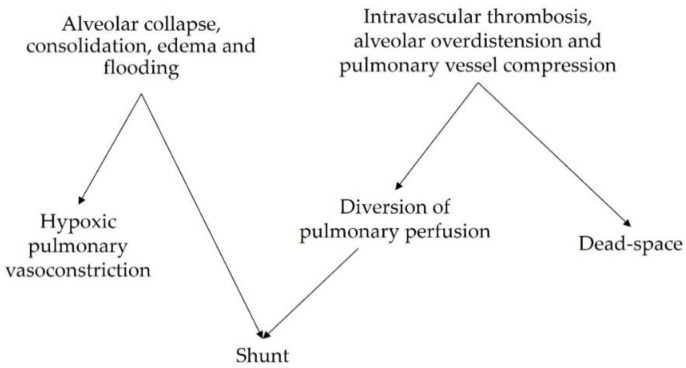
Mechanisms of *V*/*Q* mismatch and their interaction.

**Figure 2 biology-12-00067-f002:**
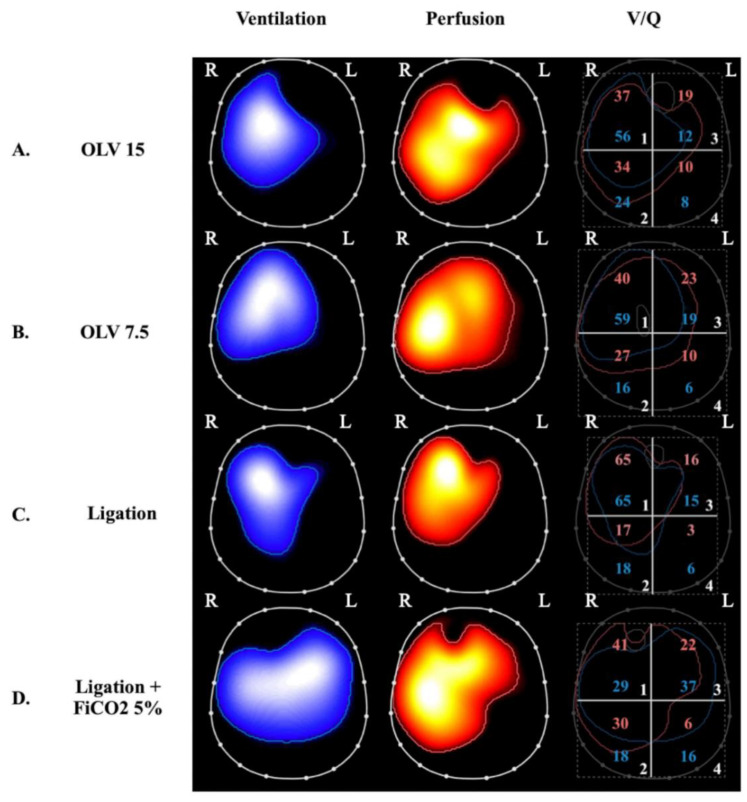
Ventilation, perfusion, and ventilation-perfusion matching via electrical impedance tomography in 4 experimental swine study groups. Images on the left display regional ventilation, middle images depict regional perfusion and were obtained by administering a hypertonic saline bolus under apneic conditions (see text), and images on the right depict ventilation-perfusion matching, which is expressed as the superposition of the ventilation and perfusion maps. The percentages of ventilation and perfusion to each of the four quadrants are annotated as blue and red numbers, respectively, on the right-side panels. The letters R and L indicate the right and left lung, respectively. (Panels **A** and **B**) were obtained during one-lung ventilation (OLV) with exclusion of the left lung and a tidal volume of 15 mL/kg (panel **A**) and 7.5 mL/kg (panel **B**) [83]. At both tidal volumes, there is no ventilation of the left lung and perfusion appears to be redistributed to the ventilated lung. OLV at higher tidal volume (panel **A**) caused bilateral lung injury (lung histological score 5 ± 2 in the right lung and 10 ± 2 in the left lung); this was compared to two-lung-ventilated controls (lung histological score 3 ± 1 in right lung and 3 ± 1 in left lung). Interestingly, lowering tidal volume to 7.5 mL/kg (panel B) attenuated inflammation and lung injury (lung histological score 3 ± 1 in the right lung and 7 ± 1 in the left lung) despite an absence of change in the overall distributions of ventilation and perfusion (ANOVA *p* ≤ 0.01 for the right lung and *p* ≤ 0.001 for the left lung). (Panels **C** and **D**) were obtained from a study of selective left pulmonary artery ligation [84]. (Panel **C**) represents ligation alone whereas (panel **D**) represents ligation + 5% inhaled CO_2_. The two groups differ significantly both for ventilation and perfusion distributions: In the ligation group, perfusion is only present in the right lung, ventilation is also diverted to the right lung, and total lung histological score was 11 ± 3. In the ligation + inhaled CO_2_ group, there is a more homogeneous distribution of ventilation and perfusion in both lungs and total lung histological score decreased to 4 ± 2 (ANOVA *p* ≤ 0.0001). The occurrence of perfusion to the ligated lung with inhaled CO_2_ is thought to transpire due to increased flow through bronchial circulation.

**Figure 3 biology-12-00067-f003:**
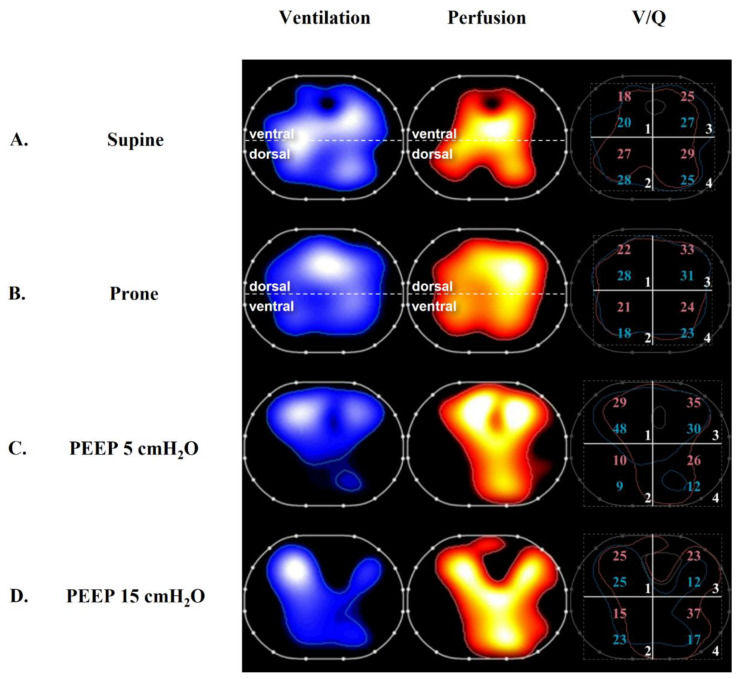
Evaluation of ventilation and perfusion maps at the bedside using electrical impedance tomography. Left-side blue panels display the regional distribution of ventilation. The red-colored middle panels display the regional distribution of perfusion (see text for details). Right-sided panels show the superposition of the contours of the ventilation and perfusion maps. The percentages of ventilation and perfusion for each of the four quadrants are annotated as blue and red numbers, respectively. (Row **A**) was obtained from a patient with COVID-19-associated acute respiratory distress syndrome (ARDS) ventilated in the supine position. The maps displayed in (row **B**) were obtained from the same patient during ventilation in the prone position, resulting in a reduction in ventilation-perfusion mismatch [60]. (Row **C**) illustrates an ARDS patient in the supine position with a set PEEP of 5 cmH_2_O. (Row **D**) was obtained from the same patient in the supine position after PEEP was increased to 15 cmH_2_O, resulting in an increase in the size of the ventilated area and improved superposition of the ventilation and perfusion maps [63].

**Table 1 biology-12-00067-t001:** *V*/*Q* mismatch as a mechanism of lung injury.

	Mechanisms of Injury	Reference
Shunt (perfused non-ventilated lung units)	Redistribution of perfusion due to hypoxic pulmonary vasoconstriction: hypo-perfused lung zones with locally decreased oxygen and nutrient delivery and lung ischemia.	[75]
	Decreased size of aerated lung with increased risk of overdistension and barotrauma in the ventilated lung.	[76,77,78]
Dead-space (ventilated non-perfused lung units)	Local alveolar hypocapnia: altered surfactant system, alveolar instability, apoptosis, and hemorrhagic infarction.	[79,80,81,82,83,84,85,86,87]
	Local bronchoconstriction with diversion of ventilation to perfused lung zones resulting in hyperventilation and hyperperfusion in diverted zones.	[88,89]

## Data Availability

Not applicable.

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
