# Peer review of "Pathophysiology and Clinical Meaning of Ventilation-Perfusion Mismatch in the Acute Respiratory Distress Syndrome"

_biology, 2022, doi:10.3390/biology12010067_

Round 1

Reviewer 1 Report

I have read the paper from Dr. Slobod and colleagues with interest. They have presented a narrative review on the topic of ventilation-perfusion mismatch in ARDS. The manuscript is well-written and easy to follow. I think the information contribute significantly to the field. I only have a few suggestions and hope to improve the manuscript further.

1. "narrative review" should be added to the title.

2. mismatch is bad but matching does not always be good, e.g. non-ventilated non-perfused units. Maybe the authors can add this to the first section.
Besides, it would be great to add a figure that illustrates the causes of various matching units and the interactions (one type of unit would turn to the other types under xx circumstance).

3. For the figures, the numbers in the third columns (both figs. 1 and 2) are too small to read. Please use a larger font and different colors that stand out from the background.

4. Figure 1. I think a little bit more explanation is required why you presented images for two OLV scenarios and what message you wanted to bring to the readers? Either in the main text or in the figure legend

5. I am curious why in panel D why perfusion is partially missing in the right lung?

6. Figure 2: Could you state in the figure legend what body position were the rows C and D?

Reviewer 2 Report

Thank you for this interesting article that emphasizes the importance of ventilation perfusion mismatch and its importance in the development of VILI and then on the use of electrical impedance tomography and the different approaches to reduce V/Q mismatch.

The article is divided to 6 main sections. I would consider combining the first two sections into an introduction of ARDS and the importance of shunt and dead space . The next two sections look the importance of these pathological processes in ARDS and then at solutions.

As this article is not directed at the intensive care community per se I think that an introduction into ARDS of an initial paragraph may be relevant, especially as there is an historical description of many aspects in the article. I would also discuss lung protective ventilation, especially if looking into survival benefit of different approaches .

Although the design lends itself to clarity I would suggest putting formulas as separate lines named and numbered so that they can be easily referred to during the reading.

I would add a detailed explanation of EIT as this is a main tool you introduce and explain the theory behind the measurement of perfusion using EIT.

I would consider looking into the survival benefits of all the suggested interventions and specifically in the context of COVID and EIT.

Figures have to be referenced (at the figure legend especially as from the text they appear to have multiple sources

Specific comments (according to sections) – many repeat the general comments

1 – add an introduction about the history and importance ARDS including inflammatory basis and VILI

1.1    – Consider adding the role of N2 as "endoskeleton"

1.2     Clarify "ventilator ratio"

2.1 modify the first sentence so that the first term the reader encounters is not "dead space" but "shunt" which is the topic of this section.
I would suggest putting equations as separate lines with relevant explanation of all variables. You elaborate very nicely about classical shunt equation yet you do not explain various other methods such as multiple FiO2.
 While TPBT is an interesting method I am not sure of its role in ARDS (maybe can be omitted altogether.

2.2 Collecting expired gasses from intubated patients cannot be defined as noninvasive.
Why not add figures explaining capnogram?
 I would be hesitant to suggest that using Harris Benedict equation (which has been shown to be inaccurate in ICU patients) is a reliable basis for calculating Vds especially as it has been done in retrospective studies without VCO2 measurement.

2.3 A more detailed explanation of EIT

3.1 again consider omitting data on TPBT

3.2 and 3.3 when separated again may distract the reader who is new to the subject as in section 2 of the article they were put together.

3.4 Requires more explanation of the technology (see comment for 2.3

4. the content of the section is hypoxic pulmonary vasoconstriction. It does not analyze VILI or the role of V/Q

4.1 Figure 1 is referred to 2 different publications (81 and 100?)  included shows VQ matching very elegantly (it refers the reader to text to understand hypertonic saline injection and measurement of perfusion – where?.) Experiment D in the figure has to be explained how can there be blood flow to the left lung if the artery was ligated?

There follows a statement on lung injury but no data on lung injury is presented

 4.2: This elaborates on the statements in table 1.

 I do not understand the relation to figure 1 as figure 1 does not signify injury and the experiments described do not lead ventilation of nonperfused lung.

Lines 332-347 (page 8) suggest that injury to the lung occurs due to regional hyperperfusion and endothelial injury. The only context that connects this paragraph to the section is that due to increased V/Q mismatch can divert ventilation to better perfused regions. I do not think that this can suggested to be related to hypoperfusion (the title of the section)

The paragraph on inhaled CO2 is very interesting .

Over all section 4 is very informative but the order and subdivision is confusing. I would suggest to start with table 1 and then elaborate on each of the statements presented. The authors bring data on ventilation and perfusion but mostly conclusions about lung injury.

Section 5:

First section succinct and clear

5.1 – clear and detailed. It should be emphasized that Literature is very old and  some new literature on covid can be added (for example Bagate F, Tuffet S, Masi P, Perier F, Razazi K, de Prost N, Carteaux G, Payen D, Mekontso Dessap A. Rescue therapy with inhaled nitric oxide and almitrine in COVID-19 patients with severe acute respiratory distress syndrome. Ann Intensive Care. 2020 Nov 4;10(1):151. doi: 10.1186/s13613-020-00769-2. PMID: 33150525;PMCID: PMC7641257.) also EIT in covid 19 (for example- Pierrakos C, van der Ven FLIM, Smit MR, Hagens LA, Paulus F, Schultz MJ, Bos LDJ. Prone Positioning Decreases Inhomogeneity and Improves Dorsal Compliance in Invasively Ventilated Spontaneously Breathing COVID-19 Patients-A Study Using Electrical Impedance Tomography. Diagnostics (Basel). 2022 Sep 21;12(10):2281.

doi: 10.3390/diagnostics12102281. PMID: 36291970; PMCID: PMC9600133.

5.2 detailed. Although reference to large studies of PEEP and survival should also be included

5.3 while previously figure 2 was referred reference 58, here it is to 55? If the authors select to include conclusions about outcome for one intervention (NO and almitrine) they should also include survival data from large trials or metanalyses of the other interventions – PEEP and Prone

Round 2

Reviewer 2 Report

I thank the authors for the detailed response